# Haptoglobin Hp1 Variant Does Not Associate with Small Vessel Disease

**DOI:** 10.3390/brainsci10010018

**Published:** 2019-12-28

**Authors:** Juha Lempiäinen, Petra Ijäs, Teemu J. Niiranen, Markku Kaste, Pekka J. Karhunen, Perttu J. Lindsberg, Timo Erkinjuntti, Susanna Melkas

**Affiliations:** 1Clinical Neurosciences, University of Helsinki, 00014 Helsinki, Finland; petra.ijas@hus.fi (P.I.); markku.kaste@hus.fi (M.K.); perttu.lindsberg@hus.fi (P.J.L.); timo.erkinjuntti@hus.fi (T.E.); susanna.melkas@hus.fi (S.M.); 2Department of Neurology, Helsinki University Hospital, Haartmaninkatu 4, 00290 Helsinki, Finland; 3Department of Public Health Solutions, National institute for Health and Welfare, Mannerheimintie 166, 00300 Helsinki, Finland; Teemu.Niiranen@thl.fi; 4Division of Medicine, Turku University Hospital and University of Turku, 20014 Turku, Finland; 5School of Medicine, University of Tampere, 33014 Tampere, Finland; pekka.karhunen@uta.fi; 6FimLab Laboratories Ltd., Tampere University Hospital Region, 33014 Tampere, Finland

**Keywords:** haptoglobins, small vessel disease (SVD), white matter lesions (WML), lacunar infarct

## Abstract

Haptoglobin (Hp) is a plasma protein that binds free hemoglobin and protects tissues from oxidative damage. An Hp2 allele has been associated with an increased risk of cardiovascular complications. On the other hand, recent studies have suggested that Hp1 allele increases risk to develop severe cerebral small vessel disease. We aimed to replicate this finding in a first-ever stroke patient cohort. Hp was genotyped by PCR and gel electrophoresis in the Helsinki Stroke Aging Memory Study in patients with DNA and magnetic resonance imaging (MRI) available (SAM; *n* = 316). Lacunar infarcts and white matter lesions (WML) classified by Fazekas grading from brain MRI were associated with Hp genotypes. As population controls, we used participants of Cardiovascular diseases—a sub study of Health 2000 Survey (*n* = 1417). In the SAM cohort, 63.0% of Hp1-1 carriers (*n* = 46), 52.5% of Hp1-2 carriers (*n* = 141) and 51.2% of Hp2-2 carriers (*n* = 129) had severe WML (*p* = 0.372). There was no difference in severe WMLs between Hp1-1 vs. Hp1-2 and Hp2-2 carriers (*p* = 0.201). In addition, 68.9% of Hp1-1 carriers (*n* = 45), 58.5% of Hp1-2 carriers (*n* = 135), and 61.8% of Hp2-2 carriers (*n* = 126) had one or more lacunar lesions (*p* = 0.472). There was no difference in the number of patients with at least one lacunar infarct between Hp1-1 vs. Hp1-2 and Hp2-2 groups (*p* = 0.322). Neither was there any difference when diabetic patients (type I and II) were examined separately. Hp1 allele is not associated with an increased risk for cerebral small vessel disease in a well-characterized Finnish stroke patient cohort.

## 1. Introduction

Haptoglobin (Hp) is a plasma protein that binds free hemoglobin (Hb). This protects tissues from heme-iron induced oxidative damage and Hb-associated nitric oxide (NO) scavenging [1,2,3]. Hp regulates vascular health by several means [4,5,6,7,8,9,10].

The Hp gene exists in two major variant alleles, Hp1 and Hp2, giving rise to three protein isoforms (Hp1-1, Hp1-2 and Hp2-2). Hp1-1 has good heme-recycling and antioxidant properties [1]. On the other hand, Hp2 allele recycles heme less efficiently and has poorer antioxidant properties [1]. Hp2-2 binds with high affinity to apolipoprotein 1A (APO1A) and apolipoprotein E (APOE) in high density lipoprotein (HDL) which interferes with reverse cholesterol transport [11,12]. The Hp2 allele has also been associated with vascular complication in diabetics [3,13,14,15]. Considering these properties, it is not surprising that our previous study showed Hp2 allele to associate with the risk of ischemic stroke in carotid stenosis patients and also with the risk of acute myocardial infarction and ischemic stroke in a general population cohort [16]. Our recent study also showed that the Hp2 allele is associated with premature ischemic cardiovascular deaths after the first-ever ischemic stroke [17].

On the other hand, several studies have reported association between Hp1 allele and cerebral white matter lesions (WML). Staals et al. reported that patients with Hp1 alleles had more lacunar lesions and extensive WML [18]. In a subsequent study [19], first ever symptomatic lacunar stroke (LACS) patients were distinguished with one or more asymptomatic lacunar lesions (*LAC+*) from those without such lesions (*LAC−*). In addition, the presence of extensive WML was defined (*WML+* or *WML−*). They found correlation between an Hp1 allele and increased risk for small vessel disease (SVD), but this correlation was only seen in the groups with milder lesions (*LAC−* and *WML−*). In another study of 152 hypertensive patients, the Hp1-1 phenotype correlated with the extent of hypertensive deep white matter (WM) damage [20]. It was hypothesized that higher blood pressure, higher serum Hp concentration, and the capability of small Hp1-1 molecules to leak through blood-brain-barrier cause damage to the perivascular brain tissue, and this would explain the increased risk of Hp1 carriers to small vessel disease (SVD) [19]. In the study of Costacou et al., the Hp1-1 allele increased risk for stroke in type 1 diabetics [21]. In a later study of the same authors, type 1 diabetic patients had more extensive WML, especially in corpus callosum, although there was no association between Hp genotype and lacunar infarcts [22].

The goal of this study was to try to replicate this earlier association of the Hp1 allele to extensive WML in a well-defined cohort of first-ever stroke patients.

## 2. Materials and Methods

### 2.1. Helsinki SAM-Study

Subjects belong to the Helsinki Stroke Aging Memory Study (SAM), which is a cross-sectional study examining the cognitive, functional, and emotional consequences of ischemic stroke. The cohort includes 486 consecutive patients aged 55 to 85 years that were admitted to Helsinki University Central Hospital between 1993 to 1995 because of suspected stroke and fulfilling the pre-set inclusion criteria. Patients underwent detailed clinical, neuropsychological, and cognitive evaluation. DNA and brain magnetic resonance imaging (MRI) were available for 316 patients. Demographics are shown in Table 1.

### 2.2. Health 2000 Survey

As population controls, we used participants of cardiovascular diseases—a sub study of the Health 2000 Survey, which is an epidemiological cross-sectional health examination survey carried out in Finland from fall 2000 to spring 2001 by the National Public Health Institute (www.terveys2000.fi). We also used this same population cohort in our previous study [17]. The overall study cohort was a two-stage stratified cluster sample (8028 persons) representing the entire Finnish population aged 30 years and above. The cardiovascular diseases sub study included 1516 persons aged 45–74 who were invited to participate in a throughout cardiovascular examinations including carotid ultrasound. Brain imaging was not included in the study protocol. The original sample size was 1867 and participation rate 82%. DNA was available from 1426 individuals.

### 2.3. Genotyping

DNA was extracted from peripheral blood leukocytes. Hp genotypes were determined by a modification of a method described by Koch and colleagues [23]: Hp alleles were amplified in two different PCR reactions and separated by agarose gel electrophoresis. Hp1 was detected using primers A 5′-GAGGGGAGCTTGCCTTTCCATTG-3′ and B 5′-GAGATTTTTGAGCCCTGGCTGGT-3′ (HpAB-PCR) and hp2 using primers C 5′-CCTGCCTCGTATTAACTGCACCAT-3′ and D 5′-CCGAGTGCTCCACATAGCCATGT-3′ (HpCD-PCR). The amplicon lengths were 1757 bp for Hp1 and 3481 bp for Hp2 in HpAB-PCR and 349 bp for Hp2 in HpCD-PCR. The HpAB and HpCD amplicons were loaded in adjacent wells and size-separated in 0.7% agarose gel with DNA size markers. Genotypes were determined by two independent readers blinded to clinical data. Discrepant samples were reanalyzed. Genotyping was successful in all samples of SAM and 97.7% of Health 2000 participants.

### 2.4. Radiology

WML were classified with four graded Fazekas classification which divides the WM in periventricular and deep WM. Each region is then given a grade.

Periventricular WM: 0 = no lesions, 1 = caps or pencil-thin lining, 2 = smooth halo, 3 = irregular periventricular signal extending into the deep WM. Deep WM: 0 = no lesions or a single punctate lesion (no), 1 = multiple punctate lesions (mild), 2 = beginning confluence of lesions (moderate), 3 = large confluent lesions (severe). The definition of WML included both periventricular and deep lesions. The lesions were classified as lacunar infarcts if they fulfilled the following criteria: they had decreased signal intensity on T1-weighted images and increased signal intensity on T2-weighted images, their size was from 3 to 20 mm in diameter, and they were subcortical and sharply marginated. This definition was used when original SAM-study was performed in 1993–1995 (before the STRIVE-criteria were defined) [24]. All radiological analyses were performed by a radiologist [25].

### 2.5. Statistical Methods

Statistical analyses were performed by IBM SPSS Statistics 22 software (Armonk, N.Y., USA). First, we compared Hp genotype frequencies and the frequency of Hp1-1 genotype compared to Hp1-2 and Hp2-2 in stroke patients with one or more lacunar infarcts or severe WML and the population controls (Table 2). Secondly, bivariate analysis of association between Hp genotypes and (1) WM changes, and (2) lacunar lesions was carried out using either Chi2 or Fisher’s exact test for nominal variables.

### 2.6. Ethical Issues

The SAM study has been approved by the Ethics committee of the Department of Clinical Neurosciences, Helsinki University Central Hospital. The Health 2000 Survey protocol was approved by the Epidemiology Ethics Committee of the Helsinki and Uusimaa hospital region, and all of the participants signed informed consents according to the Declaration of Helsinki.

## 3. Results

Subjects (*n* = 316) belong to the Helsinki SAM Study, which is a cross-sectional study examining the cognitive, functional, and emotional consequences of ischemic stroke. As population controls (*n* = 1417), we used participants of cardiovascular diseases—a sub study of the Health 2000 Survey, which is an epidemiological cross-sectional health examination survey.

### 3.1. Stroke Patients with Lacunar Infarcts or Severe WML Compared to Controls

Hp genotypes frequencies in the SAM and Health 2000 Survey are shown in Table 2. Genotype frequencies were in Hardy–Weinberg equilibrium (SAM: *p* = 0.46; Health 2000 Survey: *p* = 0.36). The genotype frequencies did not differ between patients and population controls.

First, we compared Hp genotype frequencies in stroke patients with one or more lacunar infarcts to those in the controls (Table 2). There were no significant differences. The same was true when stroke patients with either severe (Fazekas grade 3, Table 2) or moderate to severe WML (Fazekas 2-3, data not shown) were compared to controls.

### 3.2. Cerebral WML in Stroke Patients

Cerebral WML were determined in 316 patients. According to Fazekas classification, 63.0% of Hp1-1 carriers (*n* = 46), 52.5% of Hp1-2 carriers (*n* = 141) and 51.2% of Hp2-2 carriers (*n* = 129) had severe WML (Table 3). There was no statistical difference between groups. The results were the same if stroke patients with moderate to severe WML (Fazekas 2–3) were compared to patients with mild or none WML (Fazekas 0–1) (Table 3). There was also no difference when both sexes were analyzed separately.

In subgroup analysis of diabetic patients (type I and II) 22.2% of Hp1-1 carriers (*n* = 9), 41.2% of Hp1-2 carriers (*n* = 34) and 43.3% of Hp2-2 carriers (*n* = 30) had severe WML (Table 3). There was no statistical difference between groups.

### 3.3. Lacunar Infarcts in Stroke Patients

Lacunar infarcts were assessed in 306 patients. Of them, 68.9% of Hp1-1 carriers (*n* = 45), 58.5% of Hp1-2 carriers (*n* = 135) and 61.8% of Hp2-2 carriers (*n* = 126) had one or more lacunar lesions in MRI imaging (Table 4). There was no statistical difference between groups or any difference when both sexes were analyzed separately.

In subgroup analysis of diabetic patients, 66.7% of Hp1-1 carriers (*n* = 9), 48.5% of Hp1-2 carriers (*n* = 33) and 58.1% of Hp2-2 carriers (*n* = 31) had one or more lacunar lesions (Table 4). There was no statistical difference between groups.

## 4. Discussion

In this Finnish cohort of first-ever stroke patients, the Hp1 allele did not associate with the severity of SVD. Thus, we cannot replicate earlier findings reported from other populations [18,19,20,21,22]. Staals et al. defined the severity of SVD by two means [19]. First ever symptomatic lacunar stroke (LACS) patients were distinguished with one or more asymptomatic lacunar lesions (*LAC+*) from those without such lesions (*LAC−*). They also assessed MRI images for the presence of extensive WML defined according to Fazekas’ classification as T2-weighted confluent deep WM hyperintensities (score 2 or 3 on Fazekas’ scale) and/or irregular periventricular hyperintensities extending into the deep WM (score 3 on Fazekas’ scale). After that, they distinguished LACS patients with (*WML+*) from those without (*WML−*) extensive WML. In *LAC−* (*p* = 0.032) and *WML−* (*p* = 0.038) -groups, Hp1 allele frequency was significantly higher compared to controls. In *LAC+* and *WML+* -groups, there was no statistical difference between subjects and controls. In other words, a correlation between Hp1 allele and increased risk for SVD was only seen in the groups with milder lesions (LAC− and WML−).

We tried to replicate this previously documented association in a general stroke cohort by adopting similar imaging criteria, either single lacunar infarcts or extensive WML defined by a Fazekas grading system (either moderate to severe or severe alone). We compared Hp genotype frequencies in the stroke patients with one or more lacunar infarcts or extensive WML to population controls but did not note significant or consistent differences in the genotype frequencies. We also did not find an association between Hp1 allele and lacunar infarcts or extensive WML in the SAM cohort.

It should be noted that the term small vessel disease (SVD) describes different conditions including neuroimaging, pathological, and clinical features [26]. SVD is caused by pathological processes affecting the perforating cerebral arterioles, capillaries and venules, which in turn damage the cerebral WM and deep grey matter [27]. The pathogenesis of SVD is heterogeneous, probably multifactorial and not completely understood. Damage to superficial perforating arterioles leads to marked changes in cerebral WM. In contrast, damage to shorter small arteries, arising from arteria cerebri media and penetrating basal ganglia, is more likely to cause lacunar infarcts [26]. In that mentioned last, shorter small arteries are prone to atherosclerosis, whereas, in that mentioned first, superficial perforating arteries are not. This means that the risk factors causing cerebral damage may vary between these two territories [26]. WML and lacunar infarcts also lead to different kinds of clinical outcomes. Heterogeneity of SVD is a challenge and must be considered when interpreting and comparing results of different studies. It should be noted that, due to heterogeneity of SVD and its genetics, the results of our study may not be generalizable to other populations.

The strength of our study is that SAM is a well-determined, relatively large stroke cohort. Nevertheless, the sample size could be too small to detect subtle differences in Hp genotypes between patients and controls. In addition, the cohort size limits subgroup analysis. There were only 19 stroke patients age 60 or younger with a lacunar infarct, and thus we were not able to test a possible association between Hp1-1 and lacunar infarct in young patients. In younger patients, atherosclerosis is less probable etiology than in the elderly. It would be interesting to compare haptoglobin alleles between two age groups (<60 years and ≥60 years) with radiological findings compatible with SVD. Preferably at least double cohort size compared to ours would be required. Another limitation is that, in contrast to the study by Staals and colleagues, the SAM cohort was not ascertained for a lacunar stroke [19]. However, there are relatively many lacunar stroke patients in the SAM cohort due to selection/survival bias because the SAM cohort was formed three months after the index stroke when patients with the most severe strokes have died. This may have led to a decreased number of patients with a severe stroke, especially aged females with cardioembolic strokes from atrial fibrillation.

We assumed that Hp genotype frequencies in Health 2000 survey represent Hp frequencies in the Finnish population and they are in line with population frequencies reported from other Caucasian populations. Since the Health 2000 survey did not include brain imaging in the study protocol, we cannot exclude that some controls also had lacunar infarcts or WML. However, the fact that we did not detect association between lacunar infarcts or WML and Hp genotypes in the SAM cohort makes a strong association unlikely.

Ideally, future studies should include all different manifestations of SVD: in addition to lacunar infarcts and WML, cerebral microbleeds (lobar or deep), perivascular spaces, and atrophy, including the different combinations of these manifestations. Of these, deep microbleeds and perivascular spaces located in the basal ganglia would be the most probable to associate with atherosclerosis, in addition to lacunar infarcts. Analysis of haptoglobin alleles in relation to these distinct radiological findings might reveal new insights to the pathophysiology of SVD.

## 5. Conclusions

Hp 1/2 polymorphism did not associate with the severity of WML in our study. It is likely that, in the Finnish population, there is no association between Hp1 allele and SVD.

## Figures and Tables

**Table 1 brainsci-10-00018-t001:** Demographics of the Helsinki Stroke Aging Memory Study (SAM) patients.

Variable *	Hp Genotype			*p* †
	Hp1-1	Hp1-2	Hp2-2	
Genotype/allele (*n* = 316)	46 (14.6%)	141 (44.6%)	129 (40.8%)	
Sex (% males)	52.2	39.0	57.4	0.009
Age	70.1 ± 6.3	71.5 ± 8.3	70.0 ± 7.7	ns
Smoking (current or former)	65.2	47.5	50.0	ns
Smoking males	79.2	76.4	63.0	ns
Low education (< 6 years)	27.9	29.3	27.0	ns
Hypertension	47.8	49.6	49.6	ns
Diabetes mellitus	19.6	24.1	23.3	ns
Probable metabolic syndrome	19.6	22.7	20.9	ns
Total cholesterol (mmol/L)	5.4 ± 1.1	5.6 ± 1.2	5.5 ± 1.1	ns
Low density lipoprotein C (LDL-C) (mmol/L)	3.7 ± 1.1	3.8 ± 1.1	3.7 ± 1.0	ns
High density lipoprotein C (HDL-C) (mmol/L)	1.1 ± 0.3	1.2 ± 0.3	1.1 ± 0.3	ns
Myocardial infarction	17.4	17.0	20.2	ns
Atrial fibrillation	17.4	18.6	16.3	ns
Cardiac failure	21.7	21.4	21.7	ns
Peripheral arterial disease	13.0	11.3	15.5	ns

*, Mean (± standard deviation) is shown for continuous variables and percentage for nominal variables; †, Fisher’s exact test for dichotomous variables and one-way anova for continuous variables. ns = nonsignificant.

**Table 2 brainsci-10-00018-t002:** Haptoglobin genotypes in stroke patients with lacunar infarcts or severe WML compared to controls.

	Hp1-1	Hp1-2	Hp2-2	p (geno) *	p (alleles) †
All SAM patients *n* = 316	46 (14.6%)	141 (44.6%)	129 (40.8%)	0.409	0.929
SAM patients with lacunar infarcts *n* = 189	31 (16.4%)	79 (41.8%)	79 (41.8%)	0.208	0.443
SAM patients with severe WML *n* = 169	29 (17.2%)	74 (43.8%)	66 (39.1%)	0.408	0.356
Population controls *n* = 1417	203 (14.3%)	688 (48.6%)	526 (37.1%)		

*, Fisher’s exact test between the three genotypes. †, Fisher’s exact test between Hp1-1 vs. Hp1-2 and Hp2-2. Abbreviations: Hp = haptoglobin, SAM = Stroke Aging Memory-study patients, WML = White matter lesions.

**Table 3 brainsci-10-00018-t003:** White matter lesions in the Helsinki Stroke Aging Memory Study (SAM) patients.

Variable	Hp Genotype			p (geno) *	p (alleles) †
	Hp1-1	Hp1-2	Hp2-2		
All patients Fazekas 2–3	34/46 (73.9%)	103/141 (73.0%)	91/129 (70.5%)	0.892	0.860
All patients Fazekas 3 (severe)	29/46 (63.0%)	74/141 (52.5%)	66/129 (51.2%)	0.372	0.201
All diabetics Fazekas 2–3	5/9 (55.6%)	24/34 (70.6%)	17/30 (56.7%)	0.439	0.718
All diabetics Fazekas 3 (severe)	2/9 (12.3%)	14/34 (41.2%)	13/30 (43.3%)	0.600	0.303

*, Fisher’s exact test between the tree genotypes (Hp1-1, Hp1-2 and Hp2-2). †, Fisher’s exact test between Hp1-1 vs. Hp1-2 and Hp2-2. Hp = haptoglobin.

**Table 4 brainsci-10-00018-t004:** Lacunar infarcts in the Helsinki Stroke Aging Memory Study (SAM) patients.

Variable	Hp Genotype			p (geno) *	p (alleles) †
	Hp1-1	Hp1-2	Hp2-2		
All patients One or more lacunar infarcts	31/45 (68.9%)	79/135 (58.5%)	79/126 (62.7%)	0.472	0.322
Median number of lacunas (range)	1(0–6)	1(0–7)	1(0–5)	0.619	0.434
All diabetics One or more lacunar infarcts	6/9 (66.7%)	16/33 (48.5%)	18/31 (58.1%)	0.512	0.499
All diabetics Median number of lacunas (range)	1 (0–3)	0 (0–4)	1 (0–5)	0.155	0.682

*, The presence of one or more lacunas was tested by Fisher’s exact test and the median number of lacunas by Kruskal–Wallis 1-way ANOVA between the three genotypes. †, The presence of one or more lacunas was tested by Fisher’s exact test and the median number of lacunas by Mann–Whitney U-test between Hp1-1 vs. Hp1-2 and Hp2-2. Hp = haptoglobin.

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
