# Peer review of "Haptoglobin Hp1 Variant Does Not Associate with Small Vessel Disease"

_brainsci, 2019, doi:10.3390/brainsci10010018_

Round 1
Reviewer 1 Report
Several previous studies including two of the authors provided the rationale for the research designed to assess whether or not the haptoglobin Hp1 variant (of the three Hp protein isoforms), is associated with small vessel disease.
Published work cited by the authors that implicated the association of the Hp2 allele with vascular complications in diabetics, and to reduce cholesterol uptake due to Hp2’s high affinity for APO1A and APOE in HDL. One of the author’s previous studies had linked the presence of the Hp2 allele to the risk of ischemic strike in carotid stenosis patients with risk of acute myocardial infarct and ischemic stroke in a general population cohort. A more recent study of the authors linked the Hp2 allele with premature ischemic cardiovascular deaths after a first ischemic stroke. In addition, Hp1 allele had been linked with the incidence of cerebral white matter lesions and small blood vessel diseases.
The stated goals of current study were therefore to determine if Hp1 allele was associated with extensive white matter lesions in a well define cohort of first ever stroke patients. A 316 stroke patient cohort from a Helsinki Stroke Aging stroke study were genotyped. Unfortunately, in this careful study the demographic of the SAM patients revealed no obviously difference in the distributions of Hp phenotypes or diabetes or cholesterol and lipoprotein levels among the groups. Furthermore, for example there was no significant difference when sorted by Hp genotypes or with population controls, nor in stroke patients with one or more lacunar infarcts compared with controls nor with moderate or severe white matter lesions or with lacunar infarcts.
One overall conclusion drawn by the authors was that the study group was robust providing evidence that it is unlikely that a relationship does exist between lacunar infarcts or white matter lesions and Hp phenotypes. This may be so and insight may be gained by refining the study to increase the number of patients in order to investigate that certain effects might be related to age. There was 19 stroke patients less than 60 years of age and thus too few to determine any age-related effects.
Furthermore, in the Discussion section, the authors do point out that small vessel disease encompasses many different conditions. Also, ways to investigate small vessel disease including neuroimaging reveal that the pathology and clinical features of small vessel disease are heterogeneous. This is problematic and requires clarification and more detailed discussion given the negative results of this study. Also, based on the knowledge and experience that these authors have, I suggest that they include a paragraph (or edit the current Discussion section) to provide more information for ways forward for future population studies in the various groups of patients with small vessel disease. Also, the possibility that there may be age-related effects, as suggested, requires some justification and perhaps an indication of the number of patients for such a study.
Author Response
Response to Reviewer #1:
Furthermore, in the Discussion section, the authors do point out that small vessel disease encompasses many different conditions. Also, ways to investigate small vessel disease including neuroimaging reveal that the pathology and clinical features of small vessel disease are heterogeneous. This is problematic and requires clarification and more detailed discussion given the negative results of this study. Also, based on the knowledge and experience that these authors have, I suggest that they include a paragraph (or edit the current Discussion section) to provide more information for ways forward for future population studies in the various groups of patients with small vessel disease. Also, the possibility that there may be age-related effects, as suggested, requires some justification and perhaps an indication of the number of patients for such a study.
Response: We thank the reviewer for this observation. We have edited current Discussion section and added also one paragraph like suggested. It is now mentioned that; “In younger patients, atherosclerosis is less probable etiology than in the elderly. It would be interesting to compare haptoglobin alleles between two age groups (<60 years and ≥60 years) with radiological findings compatible with SVD. Preferably at least double cohort size compared to ours would be required.”. It is also mentioned that; "It should be noted that, due to heterogeneity of SVD and genetics, also results of our study might differ in some other population.”
New paragraph in Discussion tells that; "Ideally, future studies should include all different manifestations of SVD: in addition to lacunar infarcts and WML, also cerebral microbleeds (lobar or deep), perivascular spaces and atrophy, including the different combinations of these manifestations. Of these, deep microbleeds and perivascular spaces located in the basal ganglia would be the most probable to associate with atherosclerosis, in addition to lacunar infarcts. Analysis of haptoglobin alleles in relation to these distinct radiological findings might reveal new insights to the pathophysiology of SVD.”
Reviewer 2 Report
In table 1 it should beshown which test were used to compare which groups.(Chi2or Fisher test for which variables) Chi2 shoudl be described as square value, after all it comes from mathematical formula In tables you shou comparison of continous variables it should be described which test you used to compare continuos variables., Methods section- more detail about detection of Lacunar Infarcts Discousion: is it possible that found results would be different in other populations ?Author Response
Response to Reviewer #2:
1. In table 1 it should be shown which test were used to compare which groups.(Chi2 or Fisher test for which variables).
Response: We thank the reviewer for this observation. In the revised manuscript, we have added the information which statistical test was used to compare the groups in the footnote of each table.
2. Chi2 should be described as square value, after all it comes from mathematical formula.
Response: We thank the reviewer for this observation. We have analyzed all data in contingency tables by two-sided Fisher’s exact test since most of the data is unequally distributed among the cells. Fisher's Exact Test has no formal test statistic.
3. In tables you shou comparison of continous variables it should be described which test you used to compare continuos variables.
Response: We thank the reviewer for this observation. This information has been added in the footnote of each table.
4. Methods section- more detail about detection of Lacunar Infarcts.
Response: We thank the reviewer for this observation. We have now described detection of lacunar infarcts in more detail in Methods section; ”The lesions were classified as lacunar infarcts if they fulfilled the following criteria: they had decreased signal intensity on T1-weighted images and increased signal intensity on T2-weighted images, their size was from 3 to 20 mm in diameter, and they were subcortical and sharply marginated. This definition was used when original SAM-study was performed in 1993-1995 (before the STRIVE-criteria were defined) [24]. All radiological analyses were performed by a radiologist [25].”
5. Discussion: is it possible that found results would be different in other populations?
Response: We thank the reviewer for this observation. We have revised Discussion section and now it is also mentioned that these found results might be different in other populations; "It should be noted that, due to heterogeneity of SVD and genetics, also results of our study might differ in some other population."
Round 2
Reviewer 1 Report
The changes that the authors have provided in the revised manuscript have improved it.